# Investigating Hydrological Drought Characteristics in Northeastern Thailand in CMIP5 Climate Change Scenarios

Sornsawan Chatklang [ID], Piyapong Tongdeenok * and Naruemol Kaewjampa

Watershed Management and Environmental Program, Department of Conservation, Faculty of Forestry, Kasetsart University, Bangkok 10900, Thailand; sornsawan.chatkl@ku.th (S.C.); ffornmk@ku.ac.th (N.K.)
* Correspondence: fforppt@ku.ac.th

**Abstract:** In this study, we analyzed the predictions of hydrological droughts in the Lam Chiang Kri Watershed (LCKW) by using the Soil and Water Assessment Tool (SWAT) and streamflow data for 2010–2021. The objective was to assess the streamflow drought index (SDI) for 5-, 10-, 25-, and 50-year return periods (RPs) in 2029 and 2039 in two representative concentration pathway (RCP) scenarios: the moderate climate change scenario (RCP 4.5) and the high-emission scenario (RCP 8.5). The SWAT model showed high accuracy ($R^2$ = 0.82, NSE = 0.78). In RCP4.5, streamflow is projected to increase by 34.74% for 2029 and 18.74% for 2039, while in RCP8.5, a 37.06% decrease is expected for 2029 and 55.84% for 2039. A historical analysis indicated that there were frequent short-term droughts according to SDI-3 (3-month-period index), particularly from 2014 to 2015 and 2020 to 2021, and severe droughts according to SDI-6 (6-month-period index) in 2015 and 2020. The RCP8.5 projections indicate worsening drought conditions, with critical periods from April to June. A wavelet analysis showed that there is a significant risk of severe hydrological drought in the LCKW. Drought characteristic analysis indicated that high-intensity events occur with low frequency in the 50-year RP. Conversely, high-frequency droughts with lower intensity are observed in RPs of less than 50 years. The results of this study highlight an increase in severe drought risk in high emission scenarios, emphasizing the need for water management.

**Keywords:** climate change; CMIP5; hydrological drought characteristics; Streamflow Drought Index (SDI)





## 1. Introduction

Climate change exerts a significantly influence on extreme events [1], particularly droughts [2], which have long-lasting impacts on human life, the environment, industry, and the economy [3]. Hydrological droughts, characterized by below-average streamflow, are crucial issues to consider in water resource planning and management due to increasing demand and population growth [4]. As climate change continues to alter weather patterns, predicting droughts becomes essential to ensuring efficient water resource management, irrigation system operation, agricultural production, and national economic stability [5].

However, predicting hydrological droughts is challenging due to the nonstationary nature of hydrological processes influenced by climate change [5]. These challenges complicate integrated water resource management, as droughts significantly reduce available water resources; therefore, it is necessary to design strategies to balance supply and demand [6]. Moreover, climate change exacerbates drought severity on a global scale, making it imperative to evaluate hydrological drought in various climate scenarios. Human activities, such as water over-extraction and land-use changes, further influence drought characteristics [7]. The increasing frequency, duration, and intensity of droughts underscore the importance of effective monitoring. Among the various indexes used to assess the severity of these phenomena, the streamflow drought index (SDI) is widely recognized as a simple yet effective method for evaluating hydrological droughts. Numerous studies

conducted in regions such as Northern Europe [8], Australia [9], Ethiopia [10], India [11], and Turkey [12] have demonstrated the severity of droughts, highlighting the need for robust mitigation measures and a deeper understanding of the relationship between drought and climate change [13,14]. Furthermore, the Intergovernmental Panel on Climate Change (IPCC) and the Coupled Model Intercomparison Project (CMIP) have been instrumental in developing models to predict future climate changes, with CMIP5 showing enhanced performance in simulating global precipitation trends [15].

Northeastern Thailand is significantly affected by climate change, leading to decreased rainfall and streamflow, with projections indicating a 13–19% reduction in annual streamflow and shifts in seasonal patterns [16]. This decline has severely impacted the agricultural sector, especially rice farming, where yields are expected to decrease due to higher temperatures and altered rainfall patterns [15]. The Lam Chiang Kri Watershed (LCKW) is particularly vulnerable due to its geographical limitations and sandy soil, whose water-holding capacity is poor. The region experienced a 24.52% decrease in rice production during strong El Niño events, further illustrating the severe impact of drought [17]. Additionally, the region's climate variability, frequent droughts, and issues such as soil erosion and salinity exacerbate existing agricultural challenges. Consequently, sustainable water resource management and improved agricultural practices are essential to addressing these issues [18–20]. This study presents a novel perspective on the hydrological and climatic characteristics of Northeastern Thailand, a region whose unique features have been largely underexplored in existing literature. By integrating hydrological drought analysis with return period assessments under climate change scenarios, the research addresses a significant knowledge gap [21]. By focusing on Northeastern Thailand's unique challenges, the study provides valuable insights into the region's vulnerability to future droughts and contributes to the development of effective water resource management and planning strategies.

In this study, we aimed to fill the existing research gap by evaluating the impact of climate change on streamflow in the Lam Chiang Kri Watershed, assessing hydrological drought by using the streamflow drought index (SDI), and characterizing this phenomenon across different return periods in two climate change scenarios. We utilized the SWAT model, a well-established tool for simulating streamflow, based on downscaled climate projections from selected global climate models (GCMs) in two emission scenarios: RCP4.5 and RCP8.5. The analysis focuses on projections for the years 2029 and 2039, incorporating observational streamflow data for the reference period to estimate future streamflow and calculate hydrological drought.

## 2. Materials and Methods

### 2.1. Study Area

The Lam Chiang Kri Watershed (LCKW) is located in northeastern Thailand, within the Isan Plateau, and serves as the upper branch of the Mun Watershed. Covering 2959.59 square kilometers, it has an elevation range of 145 m to 593 m above sea level (Figure 1) [22]. The terrain slopes from west to east, with predominantly laterite soil, which is a type of sandy loam with poor water-holding capabilities. As a result, hydrographs in this region display sharp rising limbs, high peaks, and steep recess limbs, indicating that rainfall quickly runs off rather than soaking into the ground, leading to rapid changes in streamflow levels [23].

The LCKW experiences lower rainfall and higher temperatures than other regions in Thailand. According to data from the Thai Meteorological Department (TMD), the area receives an average annual rainfall of 947.66 mm and has an average temperature of 33 °C. The LCKW has distinct wet and dry seasons driven by monsoons [24]. The southwestern monsoon brings heavy rain from mid-May to mid-October, while the northeastern monsoon causes the dry season from mid-October to mid-February, with a transitional period occurring from mid-February to mid-May. Additionally, the Roy-al Irrigation Department of Thailand (RID) reports an average annual streamflow of 2661.51 m3/s, with 91.78% occurring during the rainy season and the remaining 8.22% in the dry season. Furthermore,

an analysis by Thailand's Land Development Department (LDD) indicated that 88.89% of the land use in the LCKW is agricultural, yet only 22.09% is irrigated. This highlights the area's high vulnerability to drought, with 45.89% of land being classified as high risk, 29.72% as moderate risk, and 24.39% as low risk. These factors underscore the region's susceptibility to drought, which significantly affects agriculture and water resources.

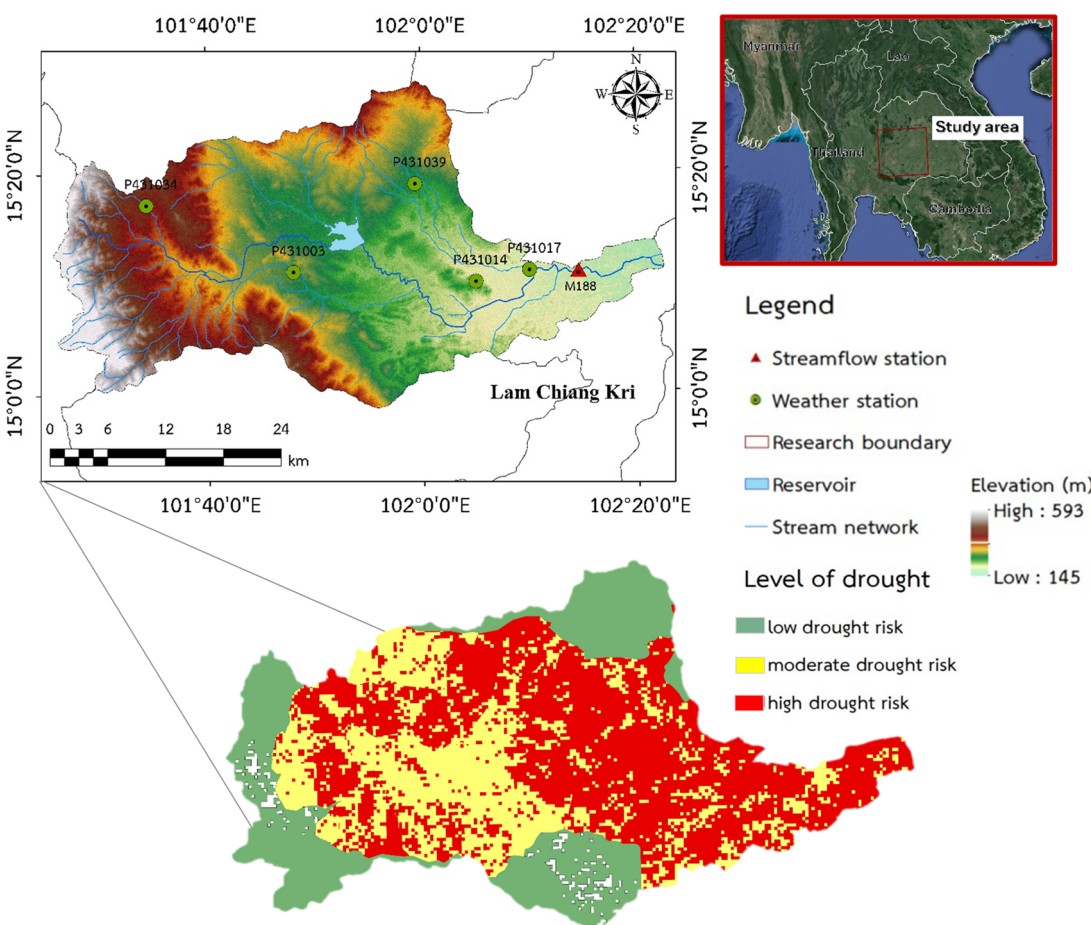

**Figure 1.** The geographical location, topographical features, historical drought patterns, and weather stations of the study area.

### 2.2. Data Collection

#### 2.2.1. Meteorological Data

Daily meteorological data, including precipitation, temperature, humidity, wind speed, and radiation, were obtained from the Thai Meteorological Department for the period from 1992 to 2022 [24]. Within the Lam Chiang Kri Watershed, data were collected from five meteorological stations: P431003 (Dan Khun Thot), P431014 (Non-Thai), P431017 (Non-Sung), P431034 (Theparak), and P431039 (Phra Thong Kham). We applied the Thiessen method to ensure an accurate representation of rainfall across the watershed. This spatial interpolation technique divides the area into polygons, with each polygon assigned the rainfall data from the nearest meteorological station. As a result, it provides a precise and reliable distribution of rainfall data across the region [25]. By doing so, the method provides a more representative average of rainfall distribution across the watershed. The processed data were then used as inputs for bias correction in climate models to evaluate past, present, and future climate scenarios.

### 2.2.2. Hydrological Data

Daily streamflow data in this study were obtained from the Irrigation Hydrology Center, Royal Irrigation Department Thailand [26]. This station is M188 (Ban Bua). The data covers April 2010–March 2021.

### 2.2.3. Topographic, Soil, and Land Use Data

The digital elevation model (DEM) was used in this study to represent the topographic condition of the study area. This DEM has a resolution of 12.5 m. These data were retrieved from the National Aeronautics and Space Administration (NASA) [27].

This study used a 2021 land use map and soil map of the LCKW, created by Thailand's Land Development Department (LDD), with reference to a spatial resolution suitable for detailed watershed analysis.

### 2.3. Methodology

In this study, we utilized daily rainfall data from five gauges of the Thai Meteorological Department (TMD), covering the historical period from 1992 to 2022. Additionally, daily streamflow data from the gauge at the M188 station, provided by the Royal Irrigation Department of Thailand (RID), were used for the historical period from 2010 to 2021. The locations of these gauges are illustrated in Figure 1, while a schematic diagram of the overall framework is shown in Figure 2. For the projected periods, two specific years were selected, the 5th year (2029) and the 15th year (2039) from the current year, to assess the impact of climate change on hydrological drought. Observational rainfall and streamflow data were crucial to calibrating and validating the Soil and Water Assessment Tool (SWAT) model version 2012, as well as calculating the baseline (2010–2021) hydrological drought index. After performing bias correction, the ability of the global climate models (GCMs) to generate streamflow for the baseline period was evaluated. The output from all selected GCMs was then analyzed, with particular emphasis on the drought index results obtained from the best-performing GCMs.

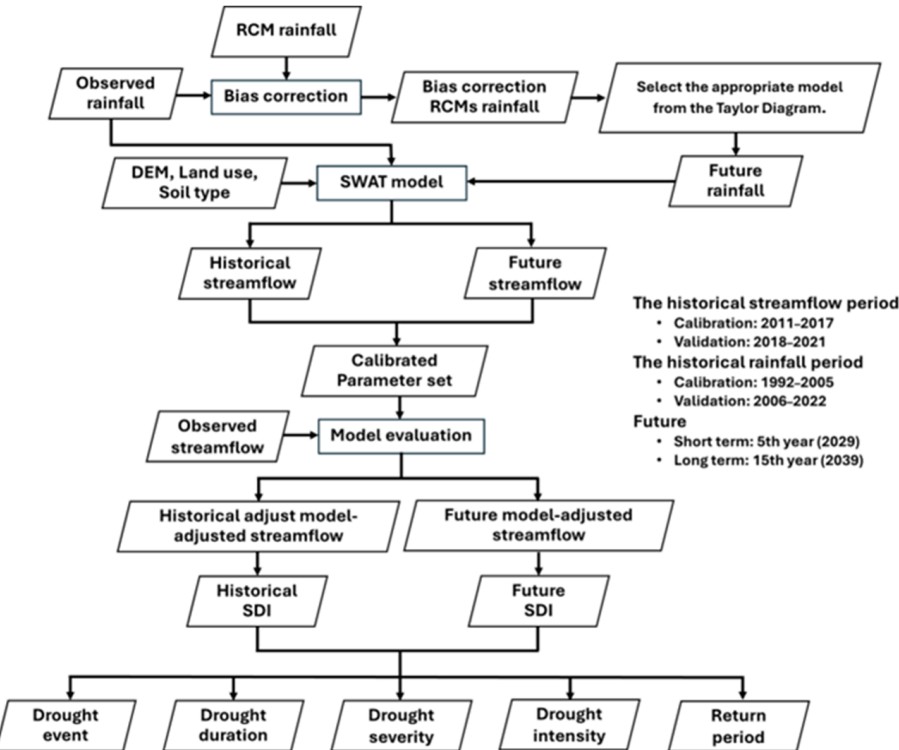

**Figure 2.** Overview of global climate model (GCM) methodology for hydrological drought assessment.

### 2.3.1. Global Climate Models (GCMs) and Climate Scenarios

In this study, we employed three well-regarded global climate models (GCMs) from the CMIP5 dataset: EC-Earth3, HadGEM2, and MPI-ESM-MR [28,29]. These models were selected for their strong capabilities in simulating historical climate variability and accurately projecting future scenarios. EC-Earth3 is particularly effective in high-resolution assessments of extreme events like floods, droughts, and heatwaves. HadGEM2 is known for its precision in modeling global warming and extreme weather conditions, making it ideal for hydrology and drought studies. MPI-ESM-MR excels in simulating complex climate interactions, such as monsoon dynamics, and is especially useful for assessing long-term drought risks in Southeast Asia [30–32]. These models were chosen for their proven ability to represent precipitation, which is a crucial variable in hydrological responses and potential future hydrological drought scenarios. These scenarios include projected frequency, duration, and severity of droughts as influenced by different climate change pathways. Specifically, CMIP5 was chosen for its ability to address the diverse and complex geography of Southeast Asia, which includes mountain ranges, major rivers, and coastal regions highly vulnerable to climate change. Its validation in numerous regional studies ensures its suitability for addressing local complexities [28]. To analyze future climate impacts, we used precipitation data from these models under two Representative Concentration Pathways (RCPs), which are greenhouse gas concentration trajectories adopted by the Intergovernmental Panel on Climate Change (IPCC). Specifically, RCP 4.5 represents a moderate climate impact scenario where radiative forcing stabilizes at 4.5 $W/m^2$ by 2100 [33], while RCP 8.5 depicts a high-emission scenario with radiative forcing reaching 8.5 $W/m^2$ by 2100, while RCP 8.5 depicts a high-emission scenario with radiative forcing reaching 8.5 $W/m^2$ by 2100. The comprehensive datasets and extensive validation of these models in similar climates ensure the reliability and robustness of our projections.

To ensure the accuracy of the projected climate data, we refined the GCM outputs through dynamic downscaling [34], improving alignment with local climate patterns and enhancing their suitability for regional impact assessments. The downscaled models were validated by comparing their outputs with observed climate data from 1992 to 2022, using statistical metrics such as correlation coefficient (r), root mean square error (RMSE), and standard deviation (SD). The models' performance was visually summarized using a Taylor diagram, allowing for comparative evaluation [35]. For our analysis of future climate impacts, we focused on two specific years, 2029 and 2039, representing the 5th and 15th years within the projected timeframe. These years were chosen to capture both near-term and mid-term climate impacts, offering insights into potential changes in hydrological patterns, particularly regarding drought conditions under different RCP scenarios. To further ensure the accuracy of these projections, we compared the models' rainfall predictions for these years with observed rainfall data from 2004 to 2022, allowing us to assess the models' reliability in projecting future climatic conditions.

### 2.3.2. SWAT Model

(1)   Model Description

The SWAT model, a semi-distributed, process-based hydrological tool developed by the United States Department of Agriculture (USDA), was employed in this study to simulate watershed processes [36], with a particular focus on assessing the impacts of climate change. The model utilized a comprehensive set of input data, including climate variables sourced from the Thai Meteorological Department, topographical data derived from a 12.5-m resolution DEM, soil characteristics provided by the Land Development Department of Thailand, and land use data reflecting both current and historical patterns. The construction of the model involved delineating the watershed, creating hydrologic response units (HRUs) based on land use, soil type, and slope, and integrating these datasets to accurately represent the hydrological processes within the watershed. Calibration and validation of the model were conducted using multi-temporal observed streamflow

data from 2010 to 2021, ensuring the model's precision and reliability in simulating the watershed's behavior under various climate scenarios [37].

In this study, the SWAT model was also utilized to delineate the watershed, divide it into sub-watersheds, and create HRUs based on land use, soil type, and slope data. The Land Development Department of Thailand classified land use into twelve primary categories, while the soil map identified forty-four distinct soil types within the study area. Additionally, the slope was categorized into five classes: flat (0–2%), sloping (2–5%), hilly (5–15%), steep (15–35%), and very steep (>35%).

A water balance equation was the basis for the SWAT model, represented as follows (Equation (1)):

$$SW_t = SW_0 + \sum \left( R_{day} - Q_{surf} - E_a - W_{seed} - Q_{gw} \right) \tag{1}$$

where $SW_0$ and $SW_t$ (mm) are the initial and final soil water on a given day, and $R_{day}$, $Q_{surf}$, $E_a$, $W_{seed}$, and $Q_{gw}$ (mm) are the rainfall, runoff, ET, water seepage to the upper soil layer, and return flow on that day, respectively.

The SWAT model used the Soil Conservation Service curve number (SCS-CN) approach to compute surface runoff in the study area. The SCS-CN equation is shown by Equation (2), as follows:

$$Q_{surf} = \frac{\left( R_{day} - I_a \right)^2}{\left( R_{day} - I_a + S \right)} \tag{2}$$

where $Q_{surf}$ is daily surface runoff (mm); $R_{day}$ is daily rainfall depth (mm); $I_a$ is the initial abstraction (mm); and $S$ is the retention parameter (mm). The retention parameter $S$ is not fixed and can be affected by factors such as slope, soil, and land-use management. Mathematically, the retention parameter can be represented as Equation (3), as follows:

$$S = 254 \times \left( \frac{100}{CN} - 1 \right) \tag{3}$$

where $S$ is the retention parameter (mm), and $CN$ is the curve number. The curve number ranges from 0 to 100, with 100 indicating no potential retention and 0 reflecting infinite potential retention [38].

(2)     Model Setup

The SWAT model calibration and validation process requires careful consideration of observation streamflow data from the M188 station. The data are divided into 80% for calibration and 20% for validation, with periods selected from 2010 to 2021. The simulation runs for 12 years, starting from 1 January 2010, to 31 December 2021. Nine sub-watersheds were created in the study area, with a threshold of 10% for land use, 10% for soil, and 10% for slope, resulting in 108 hydrologic response units.

(3)     Model Evaluation

The study used SWAT-CUP software version 5.1.6 with the Sequential Uncertainty Fitting (SUFI) algorithm to calibrate a model [39], which can handle a large number of parameters and combine sensitive analysis and improvement [37].

The model's performance was compared using three statistical performance indices: Nash and Sutcliffe Efficiency (NSE) following Equation (4) [40]; the coefficient of determination ($R^2$), following Equation (5) [41]; percent bias (PBIAS), following Equation (6) [42]; and Kling–Gupta Efficiency (KGE), following Equation (7) [43], to evaluate its daily stream-flow performance during calibration and validation phases.

$$NSE = 1 - \frac{\sum_{i=1}^{n}(Q_{obs} - Q_{sim})^2}{\sum_{i=1}^{n}(Q_{obs} - \overline{Q}_{obs})^2} \tag{4}$$

$$R^2 = \left[ \frac{\sum_{i=1}^{n} \left[ \left( Q_{obs} - \overline{Q}_{obs} \right) \left( Q_{sim} - \overline{Q}_{sim} \right) \right]}{\left[ \sum_{i=1}^{n} \left( Q_{obs} - \overline{Q}_{obs} \right)^2 \right]^{0.5} \left[ \sum_{i-1}^{n} \left( Q_{sim} - \overline{Q}_{sim} \right)^2 \right]^{0.5}} \right]^2 \tag{5}$$

$$PBIAS = \frac{\sum_{i=1}^{n} (Q_{obs} - Q_{sim})}{Q_{obs}} \times 100 \tag{6}$$

$$KGE = 1 - \sqrt{(r-1)^2 + (\alpha - 1)^2 + (\beta - 1)^2} \tag{7}$$

$Q_{obs}$ and $Q_{sim}$ represent observed and simulated values, respectively. The NSE value of the model should be more than 0.50, while the $R^2$ should be at least 0.7. PBIAS should not exceed 25 percent to be acceptable. KGE can be categorized as good (KGE $\geq$ 0.75), intermediate (0.75 > KGE $\geq$ 0.5), poor (0.5 > KGE > 0), and very poor (KGE $\leq$ 0).

2.3.3. Hydrological Drought Index

The streamflow drought index (SDI) is a key tool for assessing hydrological drought severity, where positive values indicate wetter conditions, and negative values signal the presence of drought. Calculated using monthly streamflow data, the SDI aids in managing drought and water scarcity across various time frames, encompassing both dry and wet seasons [44]. Specifically, the SDI-3, which tracks drought over a 3-month period, is particularly valuable for monitoring agricultural droughts and their impact on crops, whereas the SDI-6, calculated over 6 months, offers deeper insights into hydrological droughts that affect both surface and groundwater resources [45]. By analyzing both SDI-3 and SDI-6 together, a more comprehensive understanding of drought conditions can be achieved across both short-term and long-term time scales. Moreover, these time scale choices can be further refined depending on the specific climate and water resource focus of the study area, as detailed in Equation (8) [44].

$$SDI = \frac{V_{n,q} - V_{qm}}{S_q} \tag{8}$$

where $V_{n,q}$ represents the cumulative streamflow volume for period ($n$) and quarter ($q$) while $V_{qm}$ and $S_q$ are the mean and the standard deviation of cumulative streamflow volumes of the reference period, respectively. The classification of hydrological drought based on the SDI (Table 1) offers a detailed understanding of drought characteristics.

**Table 1.** Classification of hydrological drought based on SDI.

| State | Description | Range |
|:---:|:---:|:---:|
| 1 | No drought | $0 \leq$ SDI |
| 2 | Mild drought | $-1 \leq$ SDI $< 0$ |
| 3 | Moderate drought | $-1.5 \leq$ SDI $< -1$ |
| 4 | Severe drought | $-2 \leq$ SDI $< -1.5$ |
| 5 | Extreme drought | SDI $\leq -2$ |

The Theory of Runs (ToR) is a statistical method used to analyze drought characteristics [46], including drought event (DE), drought duration (DD), drought severity (DS), and drought intensity (DI) (Figure 3). DE is identified when the SDI value falls below a critical threshold. DD represents the duration of drought in months with negative SDI values, while DS is the sum of the absolute values of the SDI during a DE. DI can be defined as the absolute lowest value of the index (DI1) or the ratio of DS to DD in a DE (DI2).

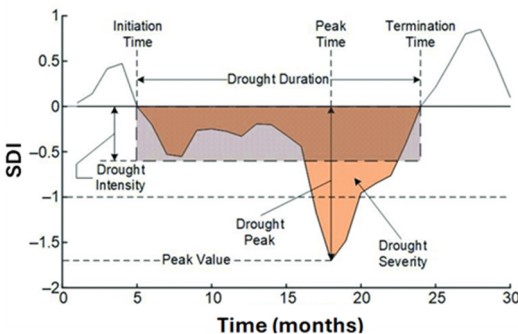

**Figure 3.** A Theory of Runs illustration of a drought event and the drought indicators [47].

2.3.4. Scenarios Analysis in Different Return Periods

The CumFreq software version 5.0. (https://www.waterlog.info/cumfreq.htm; accessed on 28 January 2024) was employed to determine the most suitable statistical distribution for characterizing drought events across various return periods (5, 10, 25, and 50 years) and time scales (3 and 6 months). CumFreq utilizes multiple probability distribution functions to analyze the input data and subsequently recommend the most appropriate distribution for drought characterization [48]. The absolute values of the streamflow drought index (SDI) were input into CumFreq to derive these distributions.

To further analyze the impact of droughts over different periods, wavelet analysis was applied to assess the variability of SDI values at different time scales and across varying return periods. This method provided deeper insights into the temporal patterns and severity of droughts under different climate scenarios. Additionally, a geostatistical approach was employed to interpolate streamflow and SDI values, which were then visualized as contour maps. These maps were generated using accurate variogram models, which are crucial for interpreting the spatial distribution of natural phenomena like drought [49]. Moreover, the relationship between streamflow and absolute SDI was explored using Surfer 21.1.158 software, enabling the creation of three-dimensional diagrams with contour lines based on the Kriging interpolation method [50].

**3. Results**

The investigation of critical hydrological droughts in the Lam Chiang Kri Watershed (LCKW) in CMIP5 climate change scenarios was divided into three key areas, i.e., the calibration and validation of the SWAT model, identification of historical drought characteristics, and assessment of climate change impacts on hydrological drought, as detailed in the following sections.

*3.1. Calibration and Validation of SWAT Model*

The SWAT model simulation for the period from 2010 to 2021, supported by data from a hydrological station within the Lam Chiang Kri Watershed, allowed us to effectively analyze the watershed's hydrological responses to varying meteorological conditions. The calibration and validation performed by using the SUFI-2 algorithm within SWAT-CUP ensured the accuracy of the streamflow patterns, significantly enhancing the reliability of the results (Table 2).

**Table 2.** Sensitivity parameters in the SWAT-CUP model of LCKW.

| Parameter | *t*-Stat | *p*-Value | Fit Value | Min Value | Max Value |
|---|---|---|---|---|---|
| 1: R__CN2.mgt | −5.48 | 0.01 | 40.395 | 35 | 100 |
| 2: R__SOL_AWC(..).sol | −2.84 | 0.04 | 0.343 | −0.2 | 0.4 |
| 3: R__ESCO.hru | 2.16 | 0.08 | 0.21525 | 0.1 | 0.35 |
| 4: V__GW_DELAY.gw | 1.82 | 0.14 | 155.5 | 0 | 500 |
| 5: R__SLSUBBSN.hru | −1.71 | 0.17 | 56.90 | 50 | 150 |

A sensitivity analysis identified five critical parameters—CN2, ESCO, SOL_AWC, GW_DELAY, and SLSUBBSN—highlighting their significant influence on the streamflow simulation. These findings provide a robust foundation for future water resource management and drought mitigation strategies in the LCKW.

The calibration and validation phases in SWAT model development depend on accurate observational streamflow data. For this study, data from the M188 station, covering the period from 2010 to 2021, were utilized. Figure 4 shows that the calibrated SWAT model for the M188 station closely matched the observational data patterns.

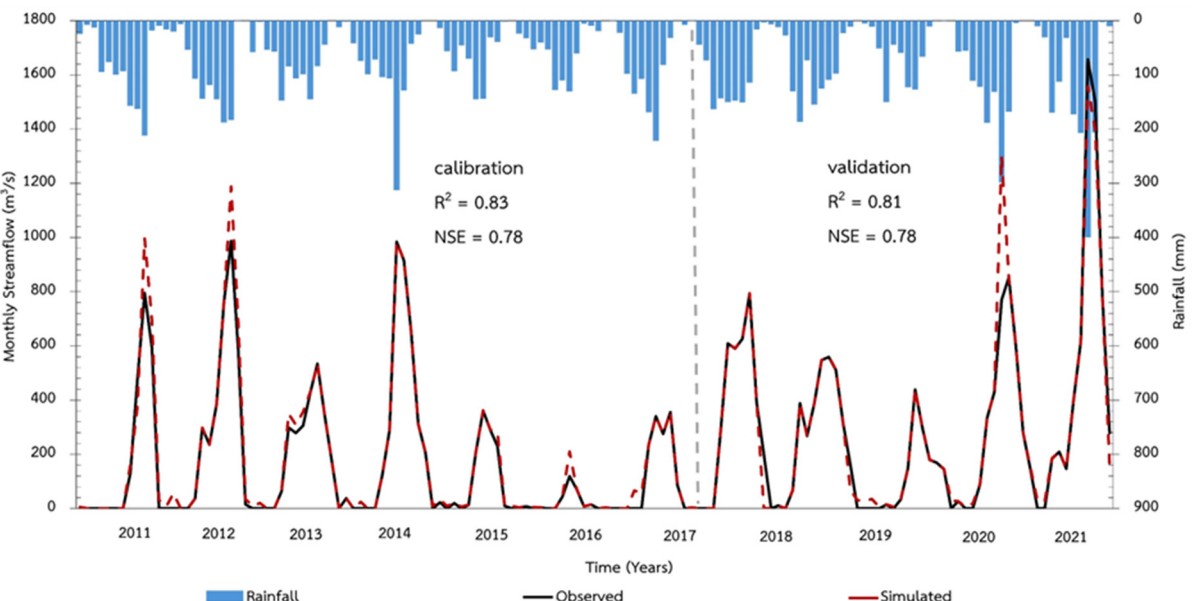

**Figure 4.** The monthly simulated and observed streamflow comparison for the M188 station during the calibration (2010–2017) and validation (2018–2021) periods. The two periods are separated by vertical dashed lines within the graphs.

For statistical evaluation, $R^2$, NSE, PBIAS, and KGE values were used. Throughout the calibration and validation periods, the streamflow station data exhibited $R^2$ and NSE values above 0.75, indicating good to very good performance. The PBIAS values were maintained below 25%, aligning with the preferred threshold. The KGE values were classified in the intermediate category, as detailed in Table 3.

**Table 3.** Statistical parameters of the SWAT model based on SWAT-CUP.

| Statistic Parameters | Calibration (2011–2017) | Validation (2018–2021) | Total (2011–2021) |
|---|---|---|---|
| $R^2$ | 0.83 | 0.81 | 0.82 |
| NSE | 0.78 | 0.78 | 0.78 |
| PBIAS | 12.0 | 28.04 | 20.02 |
| KGE | 0.64 | 0.46 | 0.55 |

### 3.2. Identification of Historical Drought Characteristics

The historical drought characteristics were calculated using the streamflow drought index (SDI) and divided into two periods: SDI-3 and SDI-6. The figure illustrates the SDI variations in the LCKW from 2010 to 2021, focusing on these two different accumulation periods. According to the 3-month accumulation period (SDI-3) analysis, the graph indicates frequent fluctuations in the SDI values (Figure 5), with significant drought periods around 2014–2015, where the mean SDI-3 value dropped to approximately −1.75. During 2020–2021, the SDI-3 values decreased to levels as low as −2.25, suggesting that short-term drought events are common and often occur annually. The 6-month accumulation period

(SDI-6) analysis showed less frequent but more severe and prolonged droughts (Figure 5). Notably, in 2015, the SDI-6 value reached approximately −2.5, and in 2020, it dropped further, to around −2.74, indicating that medium-term droughts, while less frequent, tend to be more intense and prolonged. Both the SDI-3 and SDI-6 analyses indicated that severe and extreme drought events occurred almost every year, highlighting the persistent and recurring nature of drought conditions in the region.

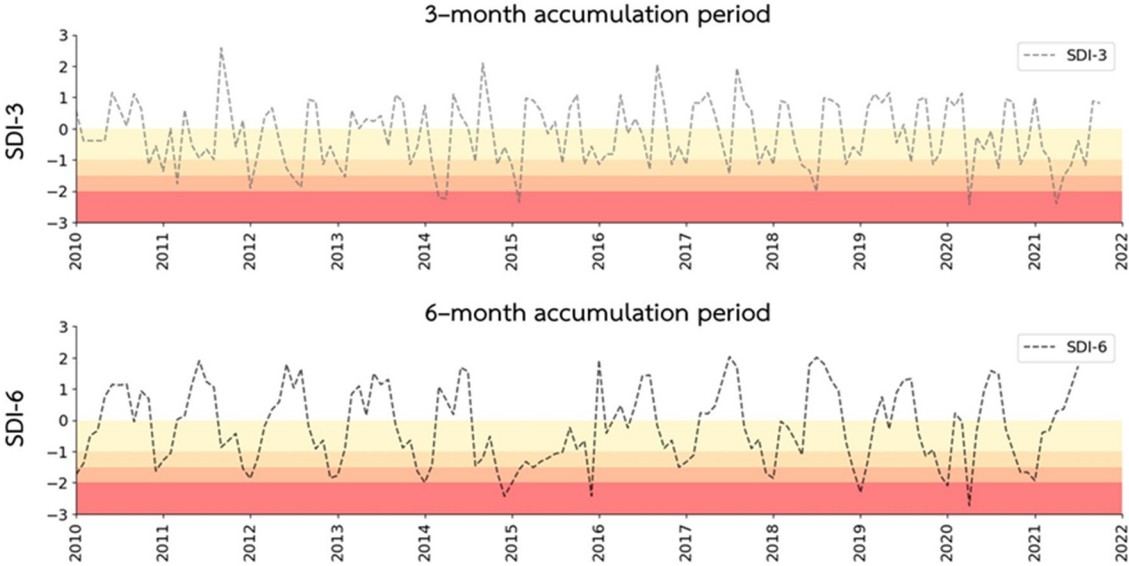

**Figure 5.** The temporal variation in the spatial averaged time series of the SDI in the LCKW at the 3- and 6-month time scales calculated based on the period of 2010–2021. The color scale from yellow to red represents mild to moderate, severe, and extreme drought categories, respectively.

The analysis indicated an average of 2.67 drought events per year, with a maximum duration of 3 months and a peak severity of −31.97. The SDI-6 analysis showed less frequent yet more severe and prolonged droughts, notably in 2015 and 2020, with an average of 1.25 drought events per year, a maximum duration of 6 months, and a peak severity of −43.04. Both indexes underscore the persistent and recurrent nature of drought conditions, highlighting the necessity for effective water resource management strategies to mitigate both short-term and medium-term drought risk. The maximum intensity values for SDI-3 were −2.44 in 2015 (DI1) and −1.35 (DI2), whereas for SDI-6, they were −2.74 (DI1) in 2020 and −1.69 (DI2) (Table 4).

**Table 4.** The historical drought characteristics in the LCKW represented by the SDIs for 3- and 6-month accumulation periods.

|  | Hydrological Drought | |
|---|---|---|
|  | SDI-3 | SDI-6 |
| Average drought event (time/year) | 2.67 | 1.25 |
| Total number of drought events (times) | 32 | 15 |
| Maximum drought duration (months) | 23 | 36 |
| Maximum drought severity | −31.97 | −43.04 |
| Maximum drought intensity based on DI1 | −2.44 | −2.74 |
| Maximum drought intensity based on DI2 | −1.35 | −1.69 |

### 3.3. Assessment of Climate Change Impacts on Hydrological Drought

3.3.1. The Selection of the Fittest GCM

In this study, we utilized daily rainfall observations from five weather stations located within the Lam Chiang Kri Watershed (LCKW) (as shown in Figure 1). The rainfall data

were aggregated and analyzed by using the Thiessen method to accurately represent the spatial distribution of rainfall across the watershed. The observational data covered the historical period from 1992 to 2022, which was divided into two distinct phases: 1992–2005 for calibration and 2006–2022 for validation. These periods were used to assess the performance of three selected CMIP5 GCMs, i.e., EC-Earth3, HadGEM2, and MPI-ESM-MR, in the RCP4.5 and RCP8.5 scenarios.

The data were processed to ensure consistency and accuracy before being used in Taylor diagram analysis. To create the Taylor diagram, data for 1992–2022 were used to assess how accurately the models predicted results compared with the observational data during that period, which were employed to assess the fit of these models. Among the models, the EC-Earth3 model (red dot) demonstrated the highest correlation with the observational data, with a correlation coefficient of 0.65, a standard deviation (SD) of 1.98, and a root mean square error (RMSE) of 3.41, making it the most accurate model. The HadGEM2 model (blue dot) also showed a strong correlation but had a higher RMSE. The MPI-ESM-MR model (green dot) exhibited moderate correlation and RMSE values (Figure 6).

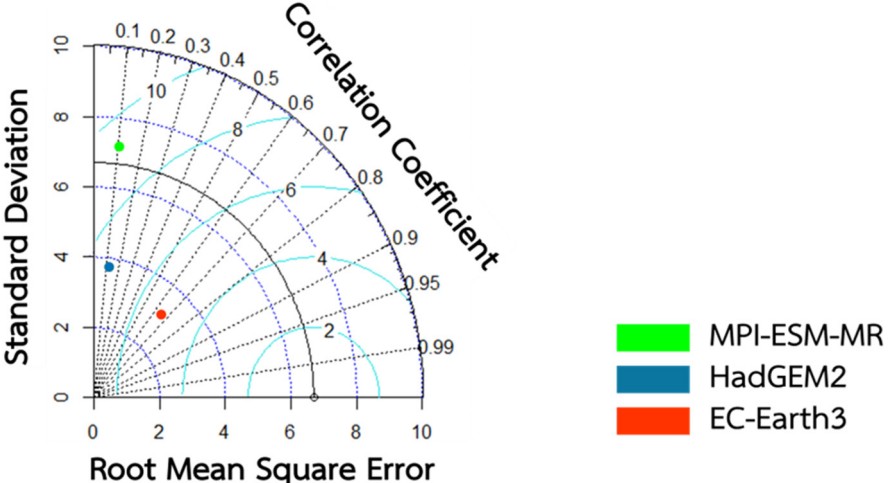

**Figure 6.** The Taylor diagram illustrates the suitability of different climate models for projecting rainfall in the LCKW.

3.3.2. Future Rainfall

The future rainfall analysis compared monthly averages from the observational periods with projections in the RCP4.5 and RCP8.5 scenarios. Figure 7 presents the EC-Earth3 model's average monthly rainfall predictions for 2029 and 2039, alongside the baseline data for 2004–2022 (Table 5), to evaluate changes in both climate scenarios.

In 2029, which represents the 5th year from the baseline reference year (2024), in the RCP4.5 scenario, the projected annual rainfall totals 1311.94 mm, with the highest monthly rainfall value occurring in September (383.03 mm) and the lowest one in January (11.46 mm). This represents a predicted increase of 38.44% in future rainfall compared with the baseline. In the RCP8.5 scenario, the projected annual rainfall is 892.97 mm, with the highest value in August (215.53 mm) and the lowest one in February (11.90 mm), indicating a decrease of 5.77% compared with the baseline.

In 2039, which represents the 15th year from the baseline reference year (2024), in the RCP4.5 scenario, the projected annual rainfall is 1101.06 mm, with the highest monthly rainfall value in August (248.36 mm) and the lowest one in January (14.80 mm), marking a 16.19% increase compared with the baseline. In the RCP8.5 scenario, the projected annual rainfall totals 805.46 mm, with the highest value in October (218.96 mm) and the lowest one in January (1.33 mm), reflecting a 15.00% decrease compared with the baseline (Table 5).

The findings indicate that there are considerable differences in the variability of monthly rainfall between the two future periods, with more substantial changes being predicted for 2039, particularly in the RCP8.5 scenario, where the most significant rainfall declines are anticipated.

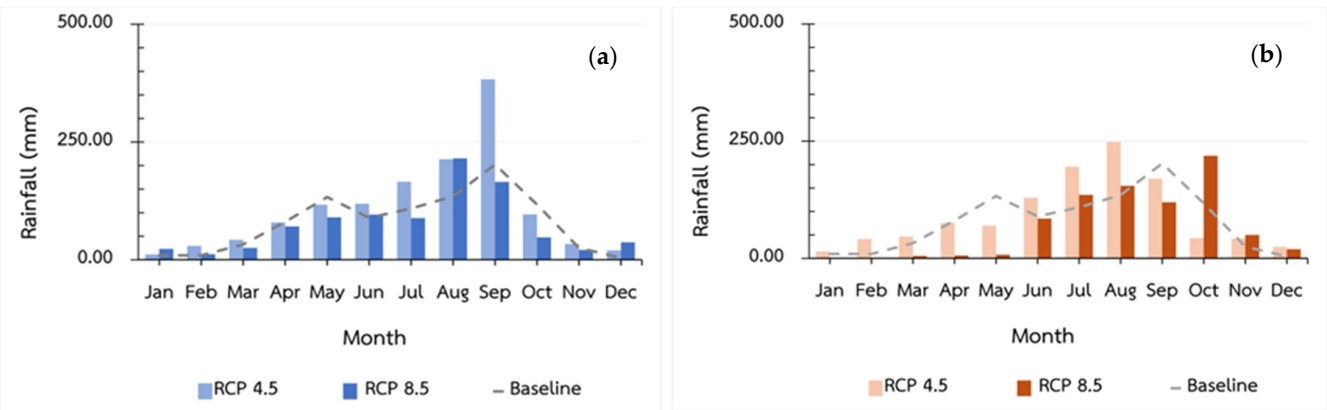

**Figure 7.** A comparison of streamflow between the baseline period (2004–2022) and the projections for 2029 (**a**) and 2039 (**b**) in the RCP4.5 and RCP8.5 future climate scenarios.

**Table 5.** Future rainfall changes in the RCP4.5 and RCP8.5 scenarios for 2029 and 2039 compared with the baseline period of 2004–2022.

| Time | Baseline (2004–2022) | RCP4.5 | | RCP8.5 | |
|---|---|---|---|---|---|
| | (mm) | (mm) | % Change | (mm) | % Change |
| 2029 | 947.64 | 1311.94 | 38.44 | 892.97 | −5.77 |
| 2039 | 947.64 | 1101.06 | 16.19 | 805.46 | −15.00 |

### 3.3.3. Future Streamflow

In this study, we analyzed variations in streamflow by comparing historical data with model simulations, focusing on mean annual streamflow in the RCP4.5 and RCP8.5 climate scenarios. Figure 8 presents the predicted average monthly streamflow for 2029 and 2039 compared with the baseline period of 2010–2021 (Table 6) to evaluate potential changes in each scenario.

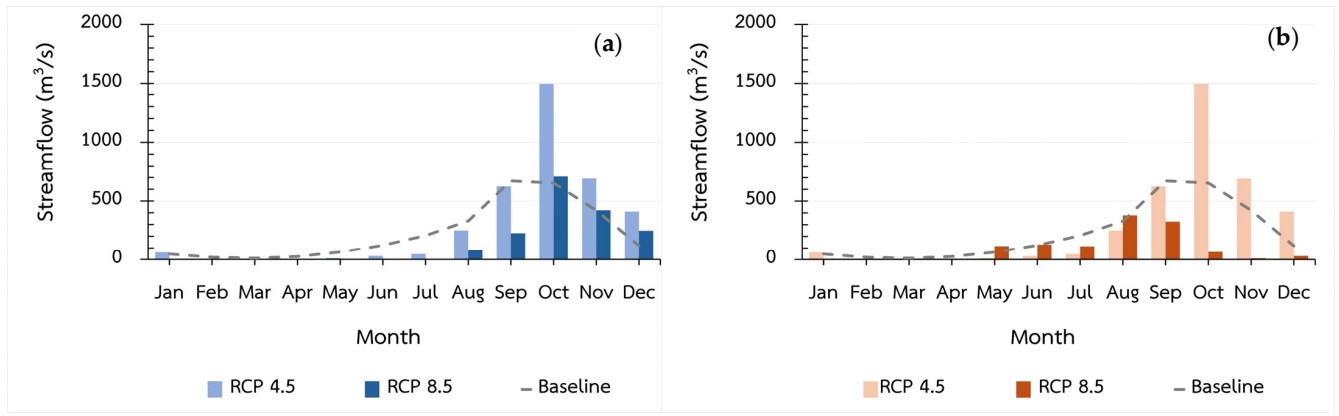

**Figure 8.** A comparison of streamflow between the baseline period (2010–2021) and the projections for 2029 (**a**) and 2039 (**b**) in the RCP4.5 and RCP8.5 future climate scenarios.

**Table 6.** A comparison of the amount of streamflow between the baseline (2010–2021) and the RCP4.5 and RCP8.5 future climate scenarios for both 2029 and 2039.

| Months | Baseline (m³/s) | | Future Streamflow (m³/s) | | | |
|---|---|---|---|---|---|---|
| | | | RCP4.5 | | RCP8.5 | |
| | | | 2029 | 2039 | 2029 | 2039 |
| | Observation | Simulation | Simulation | Simulation | Simulation | Simulation |
| January | 45.96 | 46.59 | 60.83 | 45.78 | 4.94 | 4.94 |
| February | 19.30 | 19.30 | 1.20 | 10.92 | 0.00 | 5.76 |
| March | 9.77 | 10.66 | 0.13 | 9.66 | 0.00 | 6.69 |
| April | 24.54 | 25.14 | 0.13 | 9.66 | 0.19 | 0.19 |
| May | 62.52 | 63.23 | 12.08 | 6.08 | 0.07 | 116.05 |
| June | 122.29 | 123.02 | 30.30 | 7.45 | 0.33 | 128.14 |
| July | 205.78 | 206.47 | 46.95 | 15.21 | 4.57 | 114.43 |
| August | 329.76 | 330.48 | 249.36 | 159.97 | 80.37 | 378.82 |
| September | 626.43 | 671.42 | 624.79 | 474.93 | 225.28 | 325.10 |
| October | 652.15 | 653.03 | 1495.52 | 1495.52 | 708.20 | 65.79 |
| November | 419.28 | 419.99 | 691.80 | 593.66 | 421.28 | 10.70 |
| December | 143.72 | 119.69 | 410.01 | 363.35 | 247.15 | 30.93 |
| Total runoff | 2661.51 | 2689.02 | 3623.09 | 3192.18 | 1692.38 | 1187.54 |
| Wet period (m³/s) | 2442.76 | 2492.78 | 3150.93 | 2762.47 | 1440.29 | 1139.21 |
| Dry period (m³/s) | 218.74 | 196.24 | 472.17 | 429.71 | 252.09 | 48.33 |
| (%) Percentage change | - | - | 34.74 | 18.71 | −37.06 | −55.84 |

In 2029, in the RCP4.5 scenario, the annual streamflow is projected to be 3623.09 m³/s, with the highest flow value occurring in October (1495.52 m³/s) and the lowest one in February and March (0.13 m³/s). The monsoon season is expected to contribute 86.97% (3150.93 m³/s) of the total streamflow, with the dry season contributing 13.03% (472.17 m³/s), indicating a 34.74% increase compared with the baseline period. In the RCP8.5 scenario, the streamflow is projected to decrease to 1692.38 m³/s, with the highest flow in October (708.20 m³/s) and no flow in February and March. The monsoon season is expected to contribute 85.10% (1440.29 m³/s) of the total streamflow, with the dry season contributing 14.90% (252.09 m³/s), representing a 37.06% decrease compared with the baseline period (Table 6).

In 2039, in the RCP4.5 scenario, the projected streamflow is 3192.18 m³/s, reflecting an 18.71% increase compared with the baseline. October again has the highest flow (1495.52 m³/s), and the lowest flows are expected in March and April (9.66 m³/s). The monsoon season is projected to contribute 86.54% (2762.47 m³/s) of the total streamflow, with the dry season contributing 13.46% (429.71 m³/s). Conversely, in the RCP8.5 scenario for 2039, streamflow is projected to further decrease to 1187.54 m³/s, a 55.84% decrease compared with the baseline. The peak flow is expected in August (378.82 m³/s) and the lowest value in April (0.19 m³/s). The monsoon season is expected to contribute 95.93% (1139 m³/s) of the total, with the dry season accounting for only 4.06% (48.33 m³/s).

The results highlight significant disparities in the variability in monthly streamflow between the two future time frames, with more pronounced changes being observed in 2039, especially in the RCP8.5 scenario, where the steepest declines in streamflow are projected. These changes in streamflow mirror the patterns seen in rainfall variability, indicating a strong correlation between precipitation and streamflow responses to climate change.

3.3.4. Future Hydrological Drought Characteristics

The future hydrological drought characteristics, as projected based on the SDI for the years 2029 and 2039, were analyzed in two climate scenarios, RCP4.5 and RCP8.5, based on 3-month (SDI-3) and 6-month (SDI-6) accumulation periods (Figure 9), as detailed below.

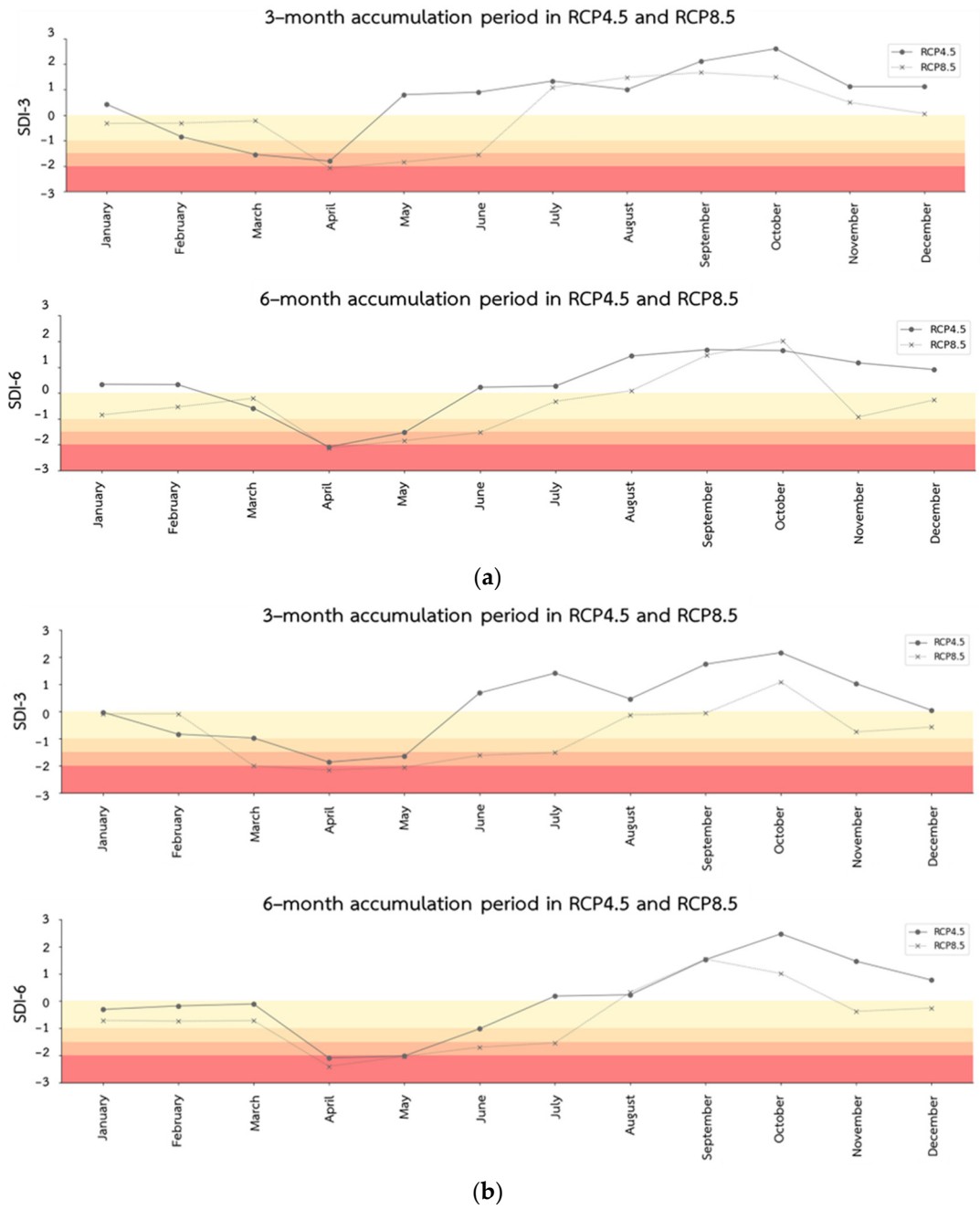

**Figure 9.** The temporal variation in the spatial averaged time series of the SDI in the LCKW at 3- and 6-month time scales calculated based on the EC-Earth3 model: 2029 (**a**); 2039 (**b**). The color scale from yellow to red represents mild to moderate, severe, and extreme drought categories, respectively.

The analysis shows that in 2029, the SDI-3 values in RCP4.5 indicate drier conditions from January to April, with improvements from May to September and a decline towards the year's end. The RCP8.5 scenario follows a similar pattern but with more severe droughts. In terms of the SDI-6 values, both scenarios exhibit dry conditions early in the year, with some recovery from April to July and a decline towards the year's end, with RCP8.5 showing more intense droughts. For 2039, both SDI-3 and SDI-6 projections in RCP4.5 and RCP8.5 suggest worsening droughts, particularly in RCP8.5, with severe droughts expected from April to July. The projections highlight an increasing trend in drought severity and frequency, especially in the higher-emission scenario (RCP8.5), emphasizing the urgent

need for effective water management and climate adaptation measures to mitigate the adverse impacts of these projected changes.

### 3.3.5. Analysis of SDI in Different Return Periods

We further investigated the relationship between streamflow and the absolute SDIs across different return periods and time scales by using wavelet analysis (Figure 10).

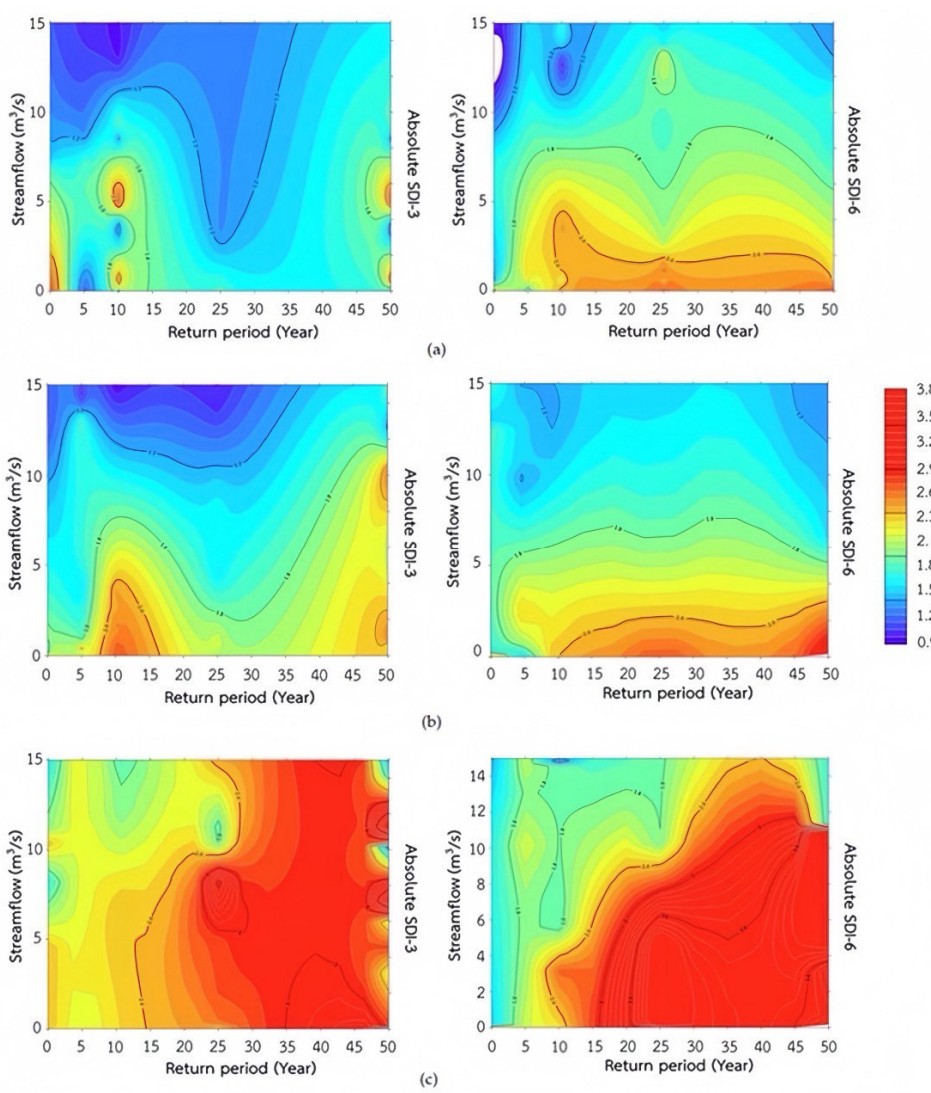

**Figure 10.** The wavelet analysis results showing the relationship among streamflow (m$^3$/s), absolute SDIs, and return periods for the baseline (**a**), RCP4.5 (**b**), and RCP8.5 (**c**). The horizontal axis represents return periods (years), and the vertical axis represents average streamflow (m$^3$/s). Bold blue lines indicate areas of low drought severity, while bold red lines mark areas of high severity. Contour lines highlight transitions between severity levels, with the color gradient further illustrating drought severity (blue for low and red for high).

For SDI-3, return periods of 2–10 years exhibit the highest drought severity, with SDI values ranging from 2.08 to 2.86, indicating that the most severe droughts occur over shorter periods. In contrast, longer return periods (over 10 years) show less intense and shorter droughts. In the RCP4.5 scenario, the most severe droughts are observed within 2–10 year return periods, rather than increasing with longer return periods. Conversely, the RCP8.5 scenario shows more severe and widespread droughts, particularly in return periods of 25–50 years, with absolute SDI values ranging from 2.26 to 4.34. Return pe-

riods of 1–25 years in RCP8.5 also show moderate drought severity across most of the watershed area.

For SDI-6, we found that return periods of 10–50 years exhibit the greatest drought severity, with absolute SDI values ranging from 2.24 to 2.74. These severe droughts cover longer periods and have a relatively long duration. In shorter return periods of 1–10 years, fewer droughts are observed. In the RCP4.5 scenario, severe droughts are primarily seen in 25–50 year return periods, with absolute SDI values ranging from 2.31 to 3.76, indicating continuous drought severity over a long period. For return periods of 0–10 years, no droughts occur in the first 5 years, and moderate droughts are observed in the 5–10 year period, with absolute SDI values ranging from 1.56 to 1.97. However, the RCP8.5 scenario shows a more extensive and severe drought distribution. In return periods of 10–50 years, high severity covers a wide area of the watershed, with absolute SDI values ranging from 2.38 to 4.38. For return periods of 1–10 years, a moderate level of drought severity covers most of the watershed area.

We found that the probability of severe hydrological drought in the LCKW is quite high due to low annual recurrence. The severity of drought varies across different periods, showing distinct behavior at different recurrence intervals. In all scenarios for SDI-3 and SDI-6, drought severity generally increases with longer return periods, except for SDI-3 in the RCP4.5 scenario, where the most severe droughts are associated with shorter return periods. Notably, SDI-6 exhibits greater severity and duration of droughts than SDI-3, as it better captures the lag between reduced rainfall and its impact on streamflow, providing a clearer indication of hydrological drought conditions.

Overall, the results of this study highlight the significant impact of higher greenhouse gas emissions on drought severity and distribution. The RCP8.5 scenario consistently indicates more severe and widespread drought conditions than the RCP4.5 scenario. The analysis underscores the importance of considering different time scales and return periods when assessing drought severity to understand the potential future impacts in varying climate change scenarios. Effective water resource management and climate mitigation efforts are crucial to addressing the increasing severity and frequency of droughts associated with higher greenhouse gas emissions.

## 4. Discussion

### 4.1. Trends in Future Rainfall

In this study, we analyzed future rainfall patterns in northeastern Thailand in two climate change scenarios, RCP4.5 and RCP8.5, by using simulations based on the EC-Earth3 model. These simulations provide valuable insights into the region's future climate, as confirmed by Pimonsree et al. (2023) [51], who proved their accuracy based on a high spatial correlation coefficient and observational rainfall data across Southeast Asia. The findings suggest a significant increase in both annual and seasonal rainfall, particularly during the rainy season from May to November [52,53]. Despite the overall increase in rainfall, the pattern of daily rainfall remains stable, which is consistent with the findings of other studies [54].

First, in the RCP4.5 scenario, researchers such as Tammadid et al. (2023) [55] and Boonwicahi et al. (2018) [56] predicted a significant increase in rainfall in northeastern Thailand by the period 2030–2035, with annual precipitation expected to increase by 13%. Additionally, for the nearby watershed area, Li et al. (2021) [57] indicated that rainfall during the wet season is projected to increase substantially in RCP4.5, which is particularly crucial for agriculture and water resource management in the region. These changes are attributed to the warming of global climate, as noted by the IPCC, which forecasted an intensified water cycle in Southeast Asia [58]. This scenario also suggests higher surface temperatures and stronger winds, potentially leading to more frequent tropical cyclones, increased rainfall, and heightened flood risk [59].

Conversely, in the RCP8.5 scenario, which assumes a higher trajectory of greenhouse gas emissions, the outlook changes significantly. Shrestha et al. (2021) [59] and Okwala

et al. (2020) [60] predicted a reduction in rainfall, with a forecasted 11% decrease by 2050 in two watershed areas close to the LCKW, which have the same climate. These nearby watersheds are relevant to our study because they share similar hydrological and meteorological characteristics, making them a useful proxy for understanding potential impacts in our primary study area. The reduction in rainfall is more pronounced during the wet seasons, which could critically impact the region's hydrology. The decrease in rainfall, combined with expanded irrigation practices and the increasing frequency and intensity of El Niño events, suggests that streamflow and water availability will be significantly affected, potentially leading to more severe drought conditions and water scarcity [61]. The results of the examination of future rainfall trends in these two distinct climate scenarios, RCP4.5 and RCP8.5, highlight significant variations in regional climate responses. This underscores the importance of considering the impacts of climate change on water management in northeastern Thailand.

*4.2. Effects on Future Streamflow*

This study assessed the impacts of climate change on streamflow, focusing on how decreases in precipitation during the rainy season contribute significantly to reductions in annual streamflow. In Southeast Asia, our findings are consistent with previous studies, such as Promping et al. (2020) [62], who projected a 3.39–6.15% decrease in streamflow from 2020 to 2050 in the RCP8.5 scenario, with rainy season flows potentially being reduced by 31–47% in the Pasak River Basin, a neighboring watershed. The Pasak River Basin shares similar climatic and hydrological characteristics with our study area, making it a valuable reference point for understanding the broader regional impacts of climate change on water resources.

Conversely, in the RCP4.5 scenario, increased rainfall could lead to significant increases in streamflow. For example, Kimmany et al. (2020) [63] reported 8% and 22% increases in dry season and annual streamflow, respectively. Similarly, Li et al. (2021) [57] projected that streamflow in the Mun River, which is the main river basin of the LCKW, could increase by 10.5%, 20.1%, and 23.2% during 2020–2093, due to high concentrations of greenhouse gases altering, cloud formation, increasing temperatures, and changing precipitation patterns [64]. Increased aerosol levels can lead to more reflective clouds [65], which are less effective in producing rain [66], while weakened atmospheric circulation further reduces storm occurrence, exacerbating water scarcity in the region [67].

However, these increases are not consistent across all regions and are influenced by various factors, including land-use changes, soil moisture retention, and evapotranspiration rates, all of which are impacted by climate change. This complexity suggests that while some regions may experience higher streamflow, others may not see corresponding increases, even with greater rainfall, underscoring the nuanced effects of climate change on hydrology [68]. Given this variability, the accuracy of hydrological model predictions becomes crucial. However, in regions with limited streamflow measurement stations, the scarcity of data can significantly hinder accurate predictions. Therefore, expanding the network of streamflow stations is essential, as it would provide more comprehensive data, thereby improving the accuracy of hydrological models and enhancing climate change projections.

*4.3. Characteristics of Hydrological Drought*

The findings suggest that climate change scenarios in RCP8.5 are likely to increase drought severity compared with historical data. Climate change is expected to alter precipitation patterns, leading to more intense and frequent drought events. These results are consistent with previous research, including the study by Satoh et al. (2022) [69], where increased drought severity due to these changes was predicted. Altered precipitation patterns are expected to exacerbate water scarcity in vulnerable areas, as highlighted by Ullah et al. (2023) [70]. While the RCP8.5 scenario projects increased drought severity, the higher emissions associated with this scenario present a greater risk, leading to more widespread

and intense droughts. These severe drought conditions could result in substantial ecological impacts, including biodiversity loss and the degradation of water resources [71].

The findings also indicate a high probability of very severe droughts in the Lam Chiang Kri Watershed (LCKW), primarily due to low annual recurrence rates. This finding is supported by Maithong et al. (2022) [45,72], who investigated the spatial distribution of drought return periods in the Mun Watershed, of which the LCKW is a branch watershed. In their study, they found increased drought severity in rivers with high streamflow over extended periods, influenced by natural variations and human activities such as dam operations and water diversions [73,74]. The diverse conditions of the selected rivers provide a comprehensive understanding of drought occurrences across different return periods. Furthermore, the choice of the SDI (e.g., SDI-3 and SDI-6) influences the results of drought behavior. SDI-3, as noted by Hasan et al. (2022) [75], is sensitive to short-term fluctuations and detects more frequent but less severe droughts [76]. In contrast, SDI-6 captures longer-term trends, reflecting more prolonged droughts [77].

While natural factors drive drought occurrences, human activities, such as land-use change and water extraction, can further exacerbate their impacts. Therefore, understanding these complexities is crucial to developing effective water management and adaptation strategies to mitigate the socioeconomic consequences of drought [78]. Although focusing on the specific years 2029 and 2039 provides valuable insights into near-term and mid-term climate impacts, this approach has its limitations. Specifically, single-year projections can effectively highlight extremes but are also susceptible to anomalies and may not fully capture long-term trends. Consequently, to enhance the reliability of future research, it would be beneficial to analyze broader periods spanning multiple decades. Such an approach would offer a more stable assessment of drought trends, thereby providing a comprehensive understanding that better informs water resource management and adaptation strategies.

*4.4. Management Implications and Future Perspectives*

Based on the results of this study, we suggest using Reservoir Operation Study (ROS) technology to optimize water storage and drainage management, particularly in the northeastern region, where unique geographical challenges complicate water management [79]. ROS technology is particularly effective in addressing these issues by enabling more precise control of water resources, ensuring that water is available when needed for agriculture, and reducing the risk of both water shortages and flooding. Additionally, farmers are encouraged to shift to less water-intensive crops and zone their cultivation based on soil and water availability. Sustainable farming practices, such as alternating wet and dry farming, using fertilizers, avoiding burning stubble and rice straw, and integrating pest management, are crucial to reducing greenhouse gas emissions and improving rice farming efficiency [80].

In the context of drought preparedness, particularly in regions such as Thailand, adopting integrated water resource management (IWRM) principles can enhance water management efficiency [81]. Developing and planting drought-resistant crops, which require less water and are more resilient to dry conditions, can significantly reduce agriculture's vulnerability to drought [82]. Empowering local communities in drought management is also essential, as community-based approaches integrate local knowledge into preparedness and response strategies.

The methodology proposed in this study can be applied globally, especially in regions such as the one studied, facilitating strategic drought management. This involves engaging stakeholders, policymakers, and water resource managers in monitoring, prediction, modeling, and disaster risk reduction. However, we acknowledge the uncertainties of this study, such as limited observational streamflow data, which affected model accuracy. Future research should incorporate changes in climate variables, land use, soil conditions, and population growth, especially in agricultural regions reliant on irrigation.

By investing in research and data collection, we can improve our ability to predict future changes and inform decision-making processes, ensuring more resilient and sustain-

able water and agricultural management practices. Finally, continued research and data collection are essential to refine our understanding of climate change impacts and develop effective adaptation and mitigation strategies.

## 5. Conclusions

Climate change impacts in Thailand, particularly in the LCKW region, are expected to vary significantly. In RCP4.5, an increase in annual and seasonal rainfall, especially during the rainy season, is projected, while RCP8.5 predicts a decrease, particularly in wet seasons, potentially leading to water scarcity. Streamflow projections show potential increases in RCP4.5, but a significant decrease is anticipated in RCP8.5.

The findings suggest that future droughts in RCP8.5 scenarios may be more intense and frequent compared with historical periods. Severe droughts are likely to occur more frequently and with greater intensity in RCP8.5. The SDI-3 analysis indicates quick-developing, short-duration droughts, whereas the SDI-6 analysis shows more widespread and prolonged drought conditions, especially in RCP8.5.

These projections underscore the urgent need for the implementation of proactive water management strategies in the LCKW, such as expanding reservoir capacity, improving irrigation efficiency, and promoting water conservation. The expected changes in rainfall and drought patterns will likely have significant socio-economic impacts, particularly on agriculture and water supply. Policymakers should prioritize investments in early warning systems, drought-resistant crops, and community-based adaptation to enhance resilience and ensure water security.

**Author Contributions:** Conceptualization, methodology, analysis, investigation, visualization, and writing—original draft, S.C.; supervision, review, and editing, P.T. and N.K. All authors have read and agreed to the published version of the manuscript.

**Funding:** This research is funded by Kasetsart University through the Graduate School Fellowship Program.

**Institutional Review Board Statement:** Not applicable.

**Informed Consent Statement:** Not applicable.

**Data Availability Statement:** The datasets, including technical and observational data, generated and/or analyzed during this study are available from the corresponding author upon reasonable request.

**Acknowledgments:** The authors are grateful to the Graduate School, Kasetsart University, for their support through the Graduate School Fellowship Program. We also thank the Thai Meteorological Department for providing climate data, including the downscaled CMIP5 GCM, and the Royal Irrigation Department for supplying hydrological data.

**Conflicts of Interest:** The authors declare no conflicts of interest.

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
