# Peer review of "Investigating Hydrological Drought Characteristics in Northeastern Thailand in CMIP5 Climate Change Scenarios"

_atmosphere, doi:10.3390/atmos15091136_

Round 1

Reviewer 1 Report

Comments and Suggestions for Authors

See below attached PDF for details.

Comments on the Quality of English Language

See the attached PDF for details.

Author Response

We sincerely appreciate the detailed and thoughtful review of our manuscript. Each of your comments has been carefully considered, leading to extensive revisions aimed at enhancing the manuscript's quality.

  1. Introduction: The introduction has been expanded to provide a more comprehensive background, including updated references to ensure both relevance and accuracy.

  2. Data Section: A dedicated section has been added to clearly describe all datasets used, ensuring transparency and reproducibility.

  3. SWAT Model Construction: This section has been detailed more thoroughly, with added explanations regarding the selection of specific climate scenarios and the calculation of drought return periods.

  4. Results: The results section has been carefully revised to correct any inconsistencies and to present the findings with greater clarity.

  5. Conclusion: The conclusion has been refined to be more concise, directly addressing the key implications of our research.

  6. Language and Formatting: We have improved the language and formatting throughout the manuscript to enhance readability and to adhere to the journal’s guidelines.

All changes have been tracked and are highlighted in the re-submitted files. Your valuable input has significantly strengthened our work, and we greatly appreciate your contributions.

For more detailed responses to each of your comments, please see the attached document.

Reviewer 2 Report

Comments and Suggestions for Authors

1. The research background and objectives should be included in the abstract. The current version of the manuscript begins with an explanation of the research methodology and the content is too lengthy. It is recommended to reorganize it.

2. The full name should be written when abbreviations first appear.

3. The latest version is CPMI6, so it needs to be explained again at the end of the second paragraph of the introduction why CPMI5 was chosen. 

4. L53-55 You have pointed out the adverse effects of human activities and climate change, which are also important factors affecting runoff. However, this article did not consider both of them, and the reasons need to be explained.

5. Delete the last sentence of the introduction.

6. The conclusion is too long and needs to be reorganized.

7. The research period of the article is too short, and the model validation period is only from 2018 to 2021. The reliability of the model needs to be discussed.

8. In the current version of the manuscript, many references are outdated. I suggest citing more from the past five years.

Author Response

We are grateful for your thorough review of our manuscript and for the thoughtful and constructive feedback provided. We appreciate the opportunity to address your comments, and we have made the necessary revisions to enhance the clarity, accuracy, and overall quality of the manuscript.

  1. Introduction: The introduction has been revised to include more recent references and to expand on the background information.

  2. Methodology and Results: We have updated these sections to improve clarity, ensuring that all procedures and findings are clearly presented.

  3. Conclusion: The conclusion has been reorganized to be more concise and focused, effectively summarizing the key contributions of the study.

  4. Methodological Rationale: Where relevant, we have explained the rationale behind our methodological choices and have provided references to support these decisions.

  5. Limitations and Future Research: We have acknowledged the limitations of the study and suggested directions for future research.

All changes are highlighted in the attached document for your review. We sincerely appreciate your feedback, which has significantly contributed to improving our work.

Round 2

Reviewer 1 Report

Comments and Suggestions for Authors

Please found below attachment for my detailed comments.

Comments on the Quality of English Language

Please found below attachment for my detailed comments.

Author Response

Dear Reviewer,

Re: Revised Manuscript Submission: "Investigating Hydrological Drought Characteristics and Different Return Periods in Semi-Arid Regions under CMIP5 Climate Change Scenarios"

Thank you for your detailed and thoughtful review of our manuscript. We have carefully considered each of your comments and have made substantial revisions to enhance the quality and clarity of the paper. Below is a summary of the major changes we have made in response to your suggestions:

  1. Introduction: We have revised the introduction to include additional relevant references and provide more comprehensive background information, specifically focusing on regional gaps in hydrological drought studies for Northeastern Thailand.
  2. Methodology: The methodologies section has been expanded with more detailed explanations of the model setup, validation process, and the rationale for selecting the CMIP5 climate scenarios. We have also incorporated additional references as suggested.
  3. Results and Conclusion: The results are now more clearly linked to the conclusions, providing better support for our findings. Minor textual and grammatical improvements have also been made throughout.

We believe that these revisions have significantly strengthened the manuscript, and we appreciate your valuable feedback that has helped improve the quality of our work.

We look forward to any further suggestions you may have and hope that this revised version meets the expectations for publication.

Sincerely,

Sornsawan Chatklang

Watershed Management and Environment Program

Faculty of Forestry, Kasetsart University

Bangkok 10900, Thailand

Reviewer 2 Report

Comments and Suggestions for Authors

Thank you for the modifications made by the author in response to the feedback. The response to the comment is also specific and scientific, and my concerns have been dispelled. Finally, please check the format of the references.

Author Response

Dear Reviewer,

Re: Revised Manuscript Submission: "Investigating Hydrological Drought Characteristics and Different Return Periods in Semi-Arid Regions under CMIP5 Climate Change Scenarios"

Thank you very much for your detailed and thoughtful review of our manuscript. We have carefully considered each of your comments and have made extensive revisions to improve the quality of the manuscript. We sincerely appreciate your comprehensive feedback, which has significantly helped to enhance the clarity and overall quality of the manuscript.

Below is a summary of the major revisions we made:

  1. Introduction: We confirmed that the introduction provides a comprehensive background and now includes all relevant references as requested.
  2. References: We ensured that all the cited references are relevant to the research, requiring no further changes in this section.
  3. Research Design and Methods: The research design was thoroughly reviewed and remains appropriate for the study objectives. The methods section has been clearly described and was not altered further as it was deemed sufficiently detailed.
  4. Results and Conclusions: We are glad to hear that the results presentation and the conclusions were found to be well-supported. No changes were required in these sections.

We believe that these revisions have significantly strengthened the manuscript. We appreciate your valuable feedback and look forward to any additional suggestions you may have.

Thank you again for your careful consideration and helpful comments.

Sincerely,

Sornsawan Chatklang

Watershed Management and Environment Program

Faculty of Forestry, Kasetsart University
